# Smart Nanofiber Mesh with Locally Sustained Drug Release Enabled Synergistic Combination Therapy for Glioblastoma

**DOI:** 10.3390/nano13030414

**Published:** 2023-01-19

**Authors:** Yinuo Li, Yoshitaka Matsumoto, Lili Chen, Yu Sugawara, Emiho Oe, Nanami Fujisawa, Mitsuhiro Ebara, Hideyuki Sakurai

**Affiliations:** 1Department of Radiation Oncology, Graduate School of Comprehensive Human Sciences, University of Tsukuba, Tsukuba 305-8575, Japan; 2Department of Radiation Oncology, Clinical Medicine, Faculty of Medicine, University of Tsukuba, Tsukuba 305-8575, Japan; 3Proton Medical Research Center, University of Tsukuba Hospital, Tsukuba 305-8576, Japan; 4Research Center for Functional Materials, National Institute for Materials Science (NIMS), 1-1 Namiki, Tsukuba 305-0044, Japan; 5Graduate School of Pure and Applied Sciences, University of Tsukuba, Tsukuba 305-0006, Japan

**Keywords:** glioblastoma, combination therapy, synergistic effect, TMZ, 17AAG, radiation therapy, radiosensitization, nanofiber

## Abstract

This study aims to propose a new treatment model for glioblastoma (GBM). The combination of chemotherapy, molecular targeted therapy and radiotherapy has been achieved in a highly simultaneous manner through the application of a safe, non-toxic, locally sustained drug-releasing composite Nanofiber mesh (NFM). The NFM consisted of biodegradable poly(ε-caprolactone) with temozolomide (TMZ) and 17-allylamino-17-demethoxygeldanamycin (17AAG), which was used in radiation treatment. TMZ and 17AAG combination showed a synergistic cytotoxicity effect in the T98G cell model. TMZ and 17AAG induced a radiation-sensitization effect, respectively. The NFM containing 17AAG or TMZ, known as 17AAG-NFM and TMZ-NFM, enabled cumulative drug release of 34.1% and 39.7% within 35 days. Moreover, 17AAG+TMZ-NFM containing both drugs revealed a synergistic effect in relation to the NFM of a single agent. When combined with radiation, 17AAG+TMZ-NFM induced in an extremely powerful cytotoxic effect. These results confirmed the application of NFM can simultaneously allow multiple treatments to T98G cells. Each modality achieved a significant synergistic effect with the other, leading to a cascading amplification of the therapeutic effect. Due to the superior advantage of sustained drug release over a long period of time, NFM has the promise of clinically addressing the challenge of high recurrence of GBM post-operatively.

## 1. Introduction

Glioblastoma (GBM) is the most common type of malignant brain tumor in adults, with a poor prognosis of a five-year overall survival rate of less than 10% [1]. Because of the diffuse and infiltrative nature of GBM, tumor cells often remain around the surgical resection site, making recurrence possible [2]. Patients with GBM have a postoperative recurrence rate of 80% to 90% [3], and there is no efficient treatment for recurrent GBM [4]. In addition to surgery, a powerful combination of treatments is necessary to eradicate GBM, which includes killing the primary tumor and the cells that have invaded the surrounding tissue.

Heat-shock protein 90 (Hsp90) is a molecular chaperone associated with the stability and function of some over-expressed signal proteins that promote the growth and survival of cancer cells [5]. Hsp90 is shown to induce resistance to radiotherapy and chemotherapy as well as contribute to local recurrence and distant metastasis [6]. The 17-allylamino-17-demethoxygeldanamycin (17AAG) is the first Hsp90 inhibitor to enter into clinical trials [7], which can bind and promote the degradation of Hsp90 client proteins [8]. Markedly, in GBM, Hsp90 client proteins are known to be disordered, including EGFR, PDGFR, Akt, and variant P53 [9]. Therefore, the molecular targeted drug 17AAG is an excellent candidate for multi-target therapy of GBM, which may provide a better prognosis. Furthermore, 17AAG has been shown to enhance the radiosensitivity of tumor cell lines, while having no influence on that of normal human fibroblast cell lines [10]. Some studies have suggested that 17AAG may downregulate Rad51 and BRCA2 protein levels, which are related proteins for DNA repair after radiation-induced damage [11,12].

Postoperative systemic temozolomide (TMZ) chemotherapy combined with radiation therapy is the standard regimen for the clinical practice of GBM [13]. Oral chemotherapeutic TMZ, an alkylating agent, can cross the blood-brain barrier (BBB). In order to form the active intermediate 5-(3-methyltriazen-1-yl) imidazole-4-carboxamide (MTIC), water molecules undergo a nucleophilic reaction. This intermediate is then converted to 4-amino-5-imidazole-carboxamide (AIC), causing DNA methylation and DNA damage [14,15]. Gastrointestinal irritation and vomiting were the main adverse effects in patients after TMZ oral administration [16]. In addition, although TMZ can cross the BBB, its ability to enter brain tissue remains limited, which makes it clinically necessary to increase the applied concentration of the drug and further raises the risk of side effects [17,18]. Thus, the delivery system for local low-dose drug release in brain tissue is highly desired to be developed to solve these problems.

To improve drug aggregation and sustained release in local tissues over long durations, various nanocarriers have been designed and synthesized based on nano-synthetic chemistry [19]. Particularly, poly-caprolactone (PCL) biomaterials have gained considerable attention due to their high biocompatibility, ease of implantability, and manipulation [20]. This biomaterial has been approved by the U.S. Food and Drug Administration for a wide range of biomedical applications because of its safety and efficiency in human tissue engineering and cancer drug delivery [21]. The PCL- nanofiber mesh (NFM), which is made from PCL biomaterials, has already shown some results in treating breast cancer and lung cancer by combining chemotherapy, radiotherapy, and hyperthermia [22,23].

Currently, electrospinning technology has been used in a wide range of fields for its unique advantages. The synthesis of nanofibrous materials for biomedicine by electrospinning has attracted considerable attention owing to their small diameter, high specific surface area, appropriate porosity, and controlled shape [24]. Most importantly, their nanofiber structure is highly similar to the structural features of the extracellular matrix (ECM) and can effectively promote cell adhesion, growth, migration, proliferation and even ECM remodeling and new tissue regeneration. In addition, numerous studies have demonstrated that electrospun nanofibers may have good antibacterial effects, which is significant for the promotion of human health [25].

This study aims to develop a new sustained-release drug-delivery system for local implantation in GBM to optimize anticancer efficacy and reduce side effects and recurrence rates by providing simultaneous and consistent multimodal effects of radiation therapy, chemotherapy, and molecular targeted therapy. As illustrated in Figure 1, the chemotherapeutic drug TMZ is integrated with 17AAG in a nano-platform NFM, which is made of electrospun PCL materials and implanted in local brain tissue lesions immediately after tumor resection by surgery. Radiation therapy was administered while the drug was consistently released locally from the NFM.

## 2. Materials and Methods

Cell culture and drugs. A human cell line derived from GBM multiforme T98G cells (RCB1954) was purchased from the RIKEN BioResource Research Center. Cells were maintained in Eagle’s minimum essential medium (E-MEM; Sigma-Aldrich, Tokyo, Japan) supplemented with 10% fetal bovine serum (FBS) and antibiotics (100 U/mL penicillin and 100 mg/mL streptomycin; Sigma-Aldrich) in a 5% CO_2_ incubator at 37 °C for further experiments. The 17AAG was obtained from Sigma-Aldrich Japan (Tokyo, Japan). TMZ was purchased from Tokyo Chemical Industry Co., Ltd. (Tokyo, Japan).

Colony formation assay. Cell survival curves were determined by colony formation assay. Cells were seeded in 6 cm dishes or T25 flasks, incubated in a 5% CO_2_ incubator at 37 °C for 48 h, and treated with drugs or irradiated with X-rays. After drug treatment or irradiation, cells were washed with PBS, separated from the dishes by 0.02% trypsin processing, diluted with a fresh medium, counted, and diluted. Cell suspensions expected to produce approximately 100 surviving cells were seeded into six cm culture dishes in triplicate and cultured for 14 days. Subsequently, the cells were fixed and stained with 10% formalin solution and 1% methylene blue solution (20% MtOH). The number of colonies was counted, with colonies consisting of 50 or more as significant colonies, and this was plotted as the cell survival rate to produce a cell survival curve.

Sensitivity and efficacy of drug combinations. Approximately 1.5 × 10^5^/3 mL of T98G cell solutions were added to six-well plates and incubated for 48 h. Subsequently, the medium was changed with the prepared medium containing different concentrations of TMZ (150 μM, 500 μM), 17AAG (100 nM, 200 nM, 350 nM), or TMZ-17AAG solution mixture, and continually the six-well plates were placed in an incubator at 37 °C and 5% CO_2_ for 24 h. Finally, the number of colonies was counted by means of a colony formation assay to calculate the cell survival rate in the case of different drug treatments. Each of the above experiments was performed at least three times. The drug combination efficacy was assessed by application of the combination index (CI) calculation [26]. The formula was as follows:Combination index (CI) = D_1_/DX_1_ + D_2_/DX_2_,(1)
where D_1_ is the half maximal inhibitory concentration (IC_50_) of 17AAG under the 17AAG+TMZ combination circumstance; DX_1_ is the IC_50_ when 17AAG is administered alone; and D_2_ is the IC_50_ of TMZ under a two-drug combination, and DX_2_ is the IC_50_ when TMZ is applied alone. It is well known that when the CI < 1, it means that TMZ and 17AAG have a synergistic effect in the T98G cell model; when CI = 1, it represents an additive effect; and when CI > 1, it indicates an antagonistic effect between TMZ and 17AAG during combination administration.

Irradiation and radiosensitization of drugs. T98G cell suspensions containing 2.5 × 10^5^/5 mL were added to T25 flasks and incubated for 48 h. Cultured T98G cells were treated with different concentrations of 17AAG (100 nM, 200 nM) or TMZ (150 μM, 500 μM) solutions for 24 h. Irradiation time was adjusted so that samples containing each concentration of 17AAG were irradiated at 0.8, 1.5, 2, 4, and 6 Gy, respectively. Samples containing TMZ were irradiated at doses of 0.8, 1.5, 3, and 6 Gy. A colony formation assay was conducted immediately after irradiation, and survival curves were obtained based on the survival rate with different irradiation and drug combinations. In this experiment, the sensitizing effect of TMZ and 17AAG drugs on radiation was determined by the sensitizer enhancement ratios (SERs). The SER is the ratio of the irradiation dose required to achieve a specific biological effect when irradiated alone and the irradiation dose required to achieve the same biological effect when irradiation is combined with the application of a radiosensitizer (such as drugs) [27]; after the application of a radiosensitizer, the irradiation dose can be reduced to achieve a specific biological effect when irradiated alone. SER > 1 indicates that the drug has a radiosensitization effect, and higher values suggest a more substantial sensitization effect.
SER = D_0_/D_C_ (for a certain survival)(2)

D_0_ is the radiation dose required for a certain survival rate (50% in this study) when irradiated alone. D_C_ is the radiation dose needed for the same survival rate when the drugs are combined.

Fabrication and characterization of nanofiber mesh. Poly(ε-caprolactone) (PCL, Mw = 80 kDa) was purchased from Tokyo Chemical Industry Co., Ltd. (Tokyo, Japan). Synthesis of PCL-NFM utilized electrospinning technology (Nanon-01A, MECC Co., Ltd., Japan). The PCL-NFM was made according to a previously published protocol [28]. The 17AAG and TMZ were dissolved in HFIP solution at a concentration of 1.72 % (*w*/*v*) and 0.007% (*w*/*v*), respectively, then the PCL solution was added prior to electrospinning, and the mixture was well stirred. A positive voltage of 20 kV was applied to the solution to allow polymer injection to form. The flow rate was set at 1.0 mL/h, the separation of the needle at 13 cm (25 gauge), and the collector plate at 25 °C and 42% humidity. The nanofibers were then collected on a plate 8 cm from the syringe needle. The morphology of PCL-NFM was investigated through a scanning electron microscopy (SEM, SU8000, Hitachi High-Technologies Corporation, Tokyo, Japan). The micrographs were recorded at 15 kV with a magnification of 1000×. The diameters of nanofibers were measured using image J.

For the mechanical analysis properties of the PCL-NFM, the ultimate tensile strength (UTS) was determined by a tensile tester (EZ-S 500N, Shimadzu, Kyoto, Japan). All experiments were conducted at an elongation rate of 5 mm/min under room temperature (*n* = 3). The sample thickness was 0.5 mm, and the width was 50 mm. The tests were carried out until the sample broke.

Furthermore, the thermal properties of the PCL-NFM were characterized using differential scanning calorimetry (DSC) (7000X, Hitachi High-Tech Science, Japan). All samples were first equilibrated at 200 °C and cooled to 0 °C. The DSC curves were obtained in the second heating run at a rate of 15 °C min^−1^ for all runs.

Long-term drug release behavior. After UV-Vis spectrum analysis of 17AAG and TMZ, the drug absorbance-wavelength standard curves of 17AAG and TMZ were detected, respectively. Each piece of the NFM (40 mg in weight) containing 1.72 % of TMZ or 0.007% of 17AAG was submerged in 5 mL of PBS solution in a glass vial and maintained at 37 °C for 35 days. At certain time intervals, 3 mL of PBS-containing drugs was withdrawn, and the same volume of fresh PBS was added. According to the wavelengths detected above, using a plate reader (Infinite 200PRO, Tecan, Männedorf, Switzerland), the fluorescence and absorbance of TMZ and 17AAG could be measured separately. Under PBS conditions, the cumulative drug release (%) of TMZ and 17AAG over 35 days could be calculated.

Cytotoxic effects with nanofiber mesh in vitro. The antitumor effect of the NFM was examined with human GBM multiforme T98G cells. T98G cell solution with a target concentration of approximately 2.5 × 10^5^/5 mL was added to the T25 flasks and incubated for 48 h. Subsequently, the medium was removed, a piece of NFM (weight 40 mg, containing 1.72% TMZ or 0.007% 17AAG) was applied, and 5 mL of fresh medium was added again as well as kept in an incubator at 37 °C and 5% CO_2_ for 24 h. After 24 h, the irradiation was carried out with an X-ray device (130 kV, 0.57 Gy/min) as described above for 158 s at an exposure dose of 1.5 Gy. Following the irradiation, cell survival and the antitumor effect under NFM administration were calculated by colony formation assay.

Statistical analysis. All the data are presented as means ± standard deviation (SD). Statistical analysis was performed using Student’s *t*-test and one-way analysis of variance (ANOVA) using GraphPad Prism software version 9.0 (GraphPad Software, San Diego, CA, USA). The *p*-value less than 0.05 (*p* < 0.05) was considered statistically significant between the results.

## 3. Results

### 3.1. Drug Sensitivity Results

The effects of 17AAG and TMZ on T98G cells were investigated individually for 24 h of treatment. Both agents impacted and decreased cell survival in a dose-dependent manner, as shown in Figure 1. The concentration ranges of 15–960 nM of 17AAG in Figure 1a and 100–800 µM of TMZ in Figure 1b were examined on T98G cells, and the IC_50_ values were determined. It was found that 17AAG and TMZ had IC_50_ concentrations of 376.4 nM and 659.7 µM for 24 h treatment of T98G cells, respectively (*p* < 0.01).

### 3.2. Synergistic Effects between TMZ and 17AAG

To determine the drug interaction between 17AAG and TMZ, 17AAG at concentrations of 100, 200, and 350 nM was administered in combination with TMZ at 150 µM and 500 µM to T98G cells treated for 24 h. The cell survival curves under the drug combination (Figure 2) were obtained according to the colony formation assay. The IC_50_ concentrations of 17AAG and TMZ under the combination of two drugs were decreased to 148.2 nM (D1) and 156.9µM (D2), respectively. From the equation in (1), as mentioned above, the CI = 0.632 < 1 means that the combination of 17AAG and TMZ, when applied to T98G, exhibited a significant drug synergistic effect. Although we did not measure the survival rate at the specific concentration of 376.4 nM (the IC_50_ concentration of 17AAG in Figure 1), it can be estimated that the IC_50_ concentration of 17AAG in Figure 2 became over 400 nM which is much bigger than 376.4 nM. We consider that this may be due to the slight variation between the actual and expected concentrations caused by human manipulation during the process of the dilutions of 17AAG. Because the units of IC_50_ for 17AAG are very small at the nanomolar scale, they would be very susceptible to errors.

### 3.3. Radiosensitization of 17AAG and TMZ

In this study, the radiation-sensitizing effects of 17AAG and TMZ were tested in the GBM T98G model, respectively. Cells treated with 100 nM and 200 nM of 17AAG were irradiated with different X-rays in Figure 3a. Cells treated with 150 µM and 500 µM of TMZ were also irradiated with different X-ray doses, shown in Figure 3b. The SERs evaluated the radiosensitization, which is calculated by the radiation dose required when treating with radiation alone and when combined with drugs to achieve a specific survival rate. According to equation (2), the SERs were 1.30 and 1.56 for treatment with 100 nM and 200 nM 17AAG, at 50% survival, and 1.18 and 1.67 for treatment with 150 µM and 500 µM. Therefore, it is suggested from our data that either 17AAG or TMZ, at least at these drug concentrations, are promising antitumor and radio-sensitizing agents for the management of GBM.

### 3.4. Synthesis and Characteristics of Nanofiber Mesh

The spatial distribution of electro-spin NFMs was evaluated through SEM. As demonstrated in the SEM images of Figure 4a, the nanofibers have uniform diameters, and the NFM samples containing 17AAG, TMZ, or the combination of two drugs showed a random reticular arrangement and similar morphology comparable to the native blank NFM. The average diameter of most NFMs was distributed between 400–700 nm, as shown in Figure 4b.

The tensile test was processed regarding the mechanical stability of the meshes. From Appendix A, the UTS of PCL-NFM was around 4.5 MPa, in the samples of thickness 0.5 mm and width 50 mm. It was demonstrated that the mechanical stability of the meshes is strong enough to implant into a tumor site.

Moreover, we further investigated the effect of drugs on the crystallinity change of nanofiber via DSC. As shown in Appendix A, the melting temperature of PCL-NFM without TMZ (59.0 °C) was almost the same as that with TMZ 10 wt.% (59.0 °C). In addition, there was no significant difference in the crystallinity change at 37 °C with and without TMZ. It revealed that the mechanical properties of the meshes do not change by adding drugs.

### 3.5. Locally Sustained Drug Release

This study aimed to demonstrate the continuous release of 17AAG and TMZ from the NFM. Figure 5 illustrates the cumulative release of 17AAG and TMZ from the NFM in PBS at pH 7.4, with a continuous release for more than 35 days. Within 24 h, the cumulative release rates of 17AAG and TMZ reached 28.2% and 35.8%, respectively. Therefore, it is expected that these drugs will be distributed in relatively high concentrations in the vicinity of the NFM shortly after their implantation into the surgical cavity of the brain. During the first 10 days, there was also a relatively explosive release of 17AAG and TMZ. However, over the following 20 days, there was a gradual and steady trend of the cumulative release of both drugs. On day 35, approximately 39.7% of the TMZ and 34.1% of 17AAG were released from TMZ-NFM and 17AAG-NFM, respectively. The cumulative release of TMZ was slightly higher than that of 17AAG, which might be due to the slightly poorer water solubility of 17AAG than that of TMZ. The above results suggest that our NFM released relatively small concentrations of 17AAG and TMZ locally and enabled a slow, low, and sustained drug release over 35 days after an initial rapid drug release within the first 24 h. A constant drug concentration maintained locally for one month is considered one of the advantages of local drug delivery methods, and this long-term drug release is attributed to the unique features of the nanofiber material.

### 3.6. Strong Cytotoxic Effects with Nanofiber Mesh Treatments

The combination of radiotherapy and chemotherapy can improve the therapeutic effect, especially the local irradiation of radiotherapy can accomplish both local treatment and alleviate the systemic side effects. In addition, the molecular targeted therapy drug Hsp90 inhibitor 17AAG has been shown to reduce the resistance of cancer cells to radiotherapy and chemotherapy. The synergistic effect between these different treatment modalities is expected to improve tumor control and allow more benefits in terms of drug dosage reduction. In the current experiment, using the NFM platform, an effort was made to investigate the effectiveness of radiotherapy in combination with the chemotherapeutic agent TMZ and the molecular targeted therapeutic agent 17AAG on human GBM T98G cells. According to the results in Figure 6, the cytotoxic effects of the single-drug NFM were significantly enhanced when combined with 1.5 Gy X-ray irradiation. The 17AAG+TMZ-NFM showed a two-drug synergistic effect in relation to the NFM of a single agent. This cell-killing effect was further intensified when applied with 1.5 Gy X-ray irradiation, leading to approximately 85% of cell death, confirming the strong cell-killing effect under the application of NFM with the simultaneous use of multiple treatments.

## 4. Discussion

GBM is one of the deadly neurological malignancies that is particularly challenging to cure because of its tumor heterogeneity, microenvironmental tumor characteristics, tumor stem-cell infiltration, and BBB properties [29]. Approximately 90% of GBM patients relapse soon after surgery within the residual cavity or margin [30]. Standard therapy consisting of surgery and radiotherapy with concomitant TMZ has improved survival from less than 1 year to about 15 months, but more effective therapies are still needed. In particular, high-intensity control of local lesions after surgery is essential to improve the survival of GBM [31]. In this experiment, we propose a novel model of implanting a nanofiber system that combines multiple therapeutic modalities and efficient antitumor treatment while also exerting the effect of long-term local drug release.

The idea of a surgically implanted cranial drug delivery tablet is not unique. The carmustine cranial tablet (GLIADEL^®^ wafer) was approved for marketing by the FDA in 1996 [32]. Carmustine (BCNU), a cytotoxic drug of the nitrosourea class, has been used in post-surgical malignant brain tumors and continues to be utilized today [33]. The therapeutic efficacy and benefits are significant, while the clinical use of the GLIADEL^®^ wafer is still controversial [34]. This is primarily because, first, the greatest weakness of BNCU is the extremely short half-life of 15 min, which requires a high clinical dosage, resulting in dose toxicity and significant side effects [35]. Second, because it is not clear how safe BCNU is in combination with other drugs, the therapeutic effect of the GLIADEL^®^ wafer with its single antitumor agent only is not very effective [36]. Third, local administration provides higher drug concentrations and fewer systemic side effects. However, previous studies on GLIADEL^®^ wafer implantation have demonstrated the risk of local complications, such as cerebral edema and hemorrhage [37]. Although there is no evidence that polyanhydride, the material from which GLIADEL^®^ is made, is toxic to the human body, it is believed that safer biomaterials than GLIADEL^®^ wafers need to be developed and utilized to reduce complications.

In our study, NFM is made of poly-caprolactone biomaterials and contains both TMZ, a chemotherapeutic agent with a better survival benefit than BCNU, and 17AAG, a molecular targeted agent. Temozolomide has been known to be more than 98% bioavailable and can cross the BBB [38]. Locally placed NFM containing TMZ further increases the concentration of the drug reaching the brain tissue, preventing the brain’s protective mechanism from expelling part of the drug out of the brain. It also has the potential to reduce gastrointestinal irritation and vomiting due to oral administration. In addition, the half-life of temozolomide, approximately 1.5 h [39], is longer than that of BCNU, which allows the concentration of the drug needed to synthesize NFM to be much lower, decreasing the probability of side effects. According to the results of this study, for T98G cells, TMZ had a significant synergistic anticancer effect with the molecular targeted drug 17AAG, which further diminished the required concentrations of both drugs and minimized the side effects produced by high doses of the drug as much as possible. These points indicate that our 17AAG+TMZ-NFM addressed some of the defects of the GLIADEL^®^ wafer and is superior to be applied in clinical practice in the future.

This experiment also considered the potential to administer radiation therapy. Postoperative TMZ combined with radiotherapy is one of the most critical tools for the clinical control of GBM [40]. D’Alessandris et al. showed that human neural stem cells were more sensitive to TMZ and radiation than glioma stem cells. A 30% tumor sphere made from glioma stem cells isolated from human GBM had an IC_50_ concentration of TMZ of 1000 μM, more than twice higher than the IC_50_ measured with human neural stem cells HNPC and NS5. Moreover, the tumor spheroids were highly resistant to a single radiation dose [41]. Therefore, in our study we focused on investigating the offsets of 17AAG to chemoresistance and radioresistance. The 17AAG is a tumor-specific, replication-dependent radiosensitizer [12]. DNA double-strand breaks (DSBs) are the leading cause of radiation-induced cell death [42]. Several studies have revealed that 17-AAG could downregulate Rad51 and BRCA2 protein levels (proteins involved in the repair of DSBs), abolishing the induction of Rad51 lesions by radiation and inhibiting the repair of double-strand breaks produced by irradiation, leading to cell death. Moreover, this effect was specific to cancer cells because 17AAG is a molecular targeted drug that binds to the Hsp90 of tumor cells with a 100-fold higher affinity than that of normal cells [43].

Multidisciplinary therapies using several modalities are becoming the standard of cancer treatment today. However, conducting radiotherapy, chemotherapy, and molecular targeted therapy simultaneously is challenging from the perspective of drug delivery as well as the size and installation standards of the devices, making it difficult to achieve multiple simultaneous treatments in reality. This problem is solved by the application of PCL-NFM. Currently, with the clinical needs, more and more nanomaterials and hydrogel materials are applied to local drug delivery. Compared with these materials, there are many outstanding advantages of PCL. It is the most-accepted synthetic polymer for medical applications because of its similarity to natural tissue components such as collagen fibers and ECM [44]. Additionally, as FDA-approved for use in biomedical devices, it has a diameter of 50–500 nm, excellent biocompatibility, and is biodegradable [45]. PCL is also cost-friendly and cheap. Toxicological tests on PCL have suggested that it is non-mutagenic and non-harmful in animal models [46]. It can also be found in Figure 6 that the cell survival rate of blank’s PCL-NFM treatment is not significantly different from the control group. PCL has a slow degradation property [47]; the most significant advantage of PCL prepared in our study is that they can release drugs in vitro for a long time (at least 35 days), which allows the maintenance of local drug therapy for a significantly longer period after cancer surgery. In addition, PCL is easy to be processed due to its good mechanical properties and is usually fabricated as an extremely thin and softer sheet [48]. This makes it compatible with the hard and soft tissues of the body, markedly mitigating the risk of poor wound healing and bleeding post-implantation. Most interestingly, a combined therapy strategy produced powerful synergistic cytotoxic effects of two drugs and a significant radiosensitization effect. These findings indicate that our NFM for GBM treatment may provide a new approach to developing localized medication delivery.

## 5. Conclusions

The 17AAG+TMZ-NFM prepared by a biodegradable poly(ε-caprolactone) material enabled the treatment of T98G cells with simultaneous chemotherapy, radiotherapy, and molecular targeted therapy through local sustained drug release, producing powerful cytotoxic effects and a significant radiosensitization effect. It may become an important approach with promising future applications for local control of GBM.

## Data Availability

Data are available from the corresponding author upon request.

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
