# Peer review of "Smart Nanofiber Mesh with Locally Sustained Drug Release Enabled Synergistic Combination Therapy for Glioblastoma"

_nanomaterials, 2023, doi:10.3390/nano13030414_

Round 1

Reviewer 1 Report

The manuscript "The smart Nanofiber mesh with locally sustained drug release 2 enabled synergistic combination therapy for glioblastoma" written by Li et al. is an interesting work on glioblastoma with an implantable nanofiber mesh loaded with 17- 48 allylamino-17-demethoxygeldanamycin and Temozolomide drug for local release of drug. The drug interactions were studied and their synergistic effects were evaluated on  T98G cells. e Radiation-sensitizing effects of these drugs were also tested. Dryg release study was performed of the nanofiber mesh with the drugs and shows a good release percent with time. The work is well described.

1) How did you determine how much drug has been loaded on the nanofibers.

Author Response

Thank you for your careful reading and helpful comments. We reply to your all comments and the reply was showed in red.

Reviewer 2 Report

Li et al. developed a PCL nanofiber mesh loading with 17AAG and TMZ for glioblastoma treatment application. Some major concerns and issues should be addressed before publication.

1. Abstract must present the research results and contribution in a better and clear way. For instance, the authors paid more attention on what they did. Instead, they should introduce more about the results. Moreover, some important results data should also be presented in this section.

2. Please state the merits and demerits of temozolomide (TMZ) and 17-allylamino-17-demethoxygeldanamycin (17AAG) compared with the other drugs in wound treatment. In other words, why are these two drugs chosen in this study?

3. The advantages and disadvantages of electrospinning should be discussed in the introduction section. Some recent works about electrospinning like 10.3390/nano11071822, and 10.1016/j.mtchem.2022.100944, should be added.

4. The characteristics of the electrospun meshes may need a bit more discussions. For example, how about the mechanical stability of the meshes? Are they strong enough? Did the addition of drugs affect the mechanical properties of electrospun meshes?

5. The SEM images of different meshes are not clear in Figure 4a, so the clear ones should be provided.

6. Does it have any significant difference in Figure 6. The statistical analysis should be performed for the biological test.

7. The grammar and writing should be improved in the whole manuscript.

Author Response

(The authors gave the same response as above.)

Reviewer 3 Report

Li et al reported a synergistic anticancer effect of radiation combined with the Hsp90 inhibitor 17AAG and the current standard chemotherapeutic drug TMZ on the human GBM cell line T98G in vitro. These drugs were loaded in a PCL-Nanofiber mesh (NFM). Their releases from the NFM were measured, and the survival of T98G cells after exposure to 17AAG-TMZ-NFM and radiation was determined with a colony formation assay, the gold-standard to assess cell toxicity after exposure to radiation. The overall aim is to implant the 17AAG-TMZ-NFM in the brain cavity after removing the primary tumor. The 17AAG and TMZ would be directly delivered into the brain to reach the remaining GBM cells without being impeded by the blood-brain barrier.

The introduction was well written but could be improved to better support the potential benefits of their 17AAG-TMZ-NFM system. For decades, nany teams have developed hydrogels or similar nanomaterials intended to be implanted in the surgical cavity where anticancer drugs were locally released. An extensive review of them is not required. Nevertheless, the authors could have summarized their major weaknesses that hinder their implementation in the clinic, and how their 17AAG-TMZ-NFM system could solve these obstacles.

A few typos need to be corrected and some abbreviations have been defined more than once. It was not possible to verify that the appropriate studies were cited since they were numbered in the text and not in the list of references.

A section describing how the results were statistically analysed should be added.

On figure 1 (a), the IC50 for 17AAG is 376.4 nM. But, on figure 2 the cell survival curve made without TMZ suggests an IC50 of about 490 nM. How can this apparent discrepancy be explained?

More details would be appreciated on the fabrication of the nanofiber mesh. A reference reporting its fabrication should be added.

Line 210: Replace “different X-rays” by “different X-ray doses”. Idem on line 211.

Line 217: Replace “at least at several drug concentrations” by “at least at these drug concentrations”.

The authors are invited to add on Figure 3 at which radiation dose the cell survival was significantly different from X-ray alone vs X-ray + drug.

Figure 3: Since the LD50 was used to compare the different treatment conditions, it would be appropriate to change the scale of the surviving fraction axis to better appreciate the different between these cell survival curves at their respective LD50.

Figure 6. Authors are encouraged to perform statistical analysis to determine which groups are significantly different and add the appropriate number of "*" to the figure based on the calculated P value.

Continuous release of 17AAG and TMZ from the NFM: The authors should mention that about 80% of 17AAG and TMZ were released from the NFM after 24 hours, which was followed by a slow and continuous release of these drugs. Therefore, it is expected that a high concentration of these drugs will be distributed in the vicinity of the NFM soon after its implantation into the surgical cavity of the brain. Therefore, their conclusion that the PCL-Nanofiber mesh provides sustained drug release is not fully supported by their results. The initial rapid release of the drug is actually followed by a slow, low and sustained release.

The GBM cells T98G are derived from human which involves that they have to be implanted in immunodeficient animals. Although this choice is defensible, it must be kept in mind that radiotherapy works in concert with the immune system, in particular the CD8+ lymphocytes and Natural Killer, to eliminate cancer cells. A syngeneic model with immunocompetent animals would thus be closer to the clinical reality.

The authors should be aware of the work done by the team of D’Alessandris et al (Neuro Oncol. 19, 1097-1108, 2017). They have shown that 30% of tumorspheres made with glioma stemlike cells isolated from human GBM tumors have a half-maximal inhibitory concentration (IC50) for TMZ of 1000 μM, which was more than 2-fold higher than the IC50 measured with the human neural stem cells HNPC and NS5. In addition, another ~ 40% of patient-derived tumorspheres have shown a IC50 similar to the human neural stem cells. The dose of radiation lethal for 50% of the cancer cells (LD50) was also determined. These tumorspheres were very resistant to single radiation dose, since the LD50 of 43.6% of them was greater than 60 Gy, which corresponds to the dose accumulated in the brain during fractional radiotherapy. The LD50 values of the tumorspheres were far superior to those of human neural stem cells HNPC and NS5, which ranged between 10 and 12 Gy. They conclude that human neural stem cells are frequently more sensitive to TMZ and radiation than glioma stemlike cells.

Author Response

(The authors gave the same response as above.)

Round 2

Reviewer 2 Report

The reviewer's comments have been addressed.